# Neuroblastoma: Tumor Biology and Its Implications for Staging and Treatment

**DOI:** 10.3390/children6010012

**Published:** 2019-01-17

**Authors:** Kyle J. Van Arendonk, Dai H. Chung

**Affiliations:** 1Department of Surgery, Children’s Hospital of Wisconsin, Medical College of Wisconsin, Milwaukee, WI 53226, USA; 2Department of Surgery, Children’s Medical Center Dallas, UT Southwestern Medical Center, Dallas, TX 75235, USA; Dai.Chung@UTSouthwestern.edu

**Keywords:** neuroblastoma, oncology, pediatric, pathology, surgery

## Abstract

Neuroblastoma, the most common extracranial solid tumor of childhood, has widely variable outcomes dependent on the specific biology of the tumor. In this review, current biologic principles that are used to stratify risk and guide treatment algorithms are discussed. The role for surgical resection in neuroblastoma is also reviewed, including the indications and timing of surgery within the greater treatment plan.

## 1. Introduction

Neuroblastoma is the most common extracranial solid tumor of childhood and represents a neoplastic expansion of neural crest cells in the developing sympathetic nervous system. The primary tumor originates anywhere along the sympathetic chain but most frequently arises from the adrenal gland. The prognosis for neuroblastoma varies widely, from tumors that spontaneously regress and require no intervention to those that present widely metastatic and resistant to therapy with resulting high mortality. This disparate prognosis is largely dependent on tumor biology, and extensive research has been completed identifying tumor characteristics that associate with aggressive tumor behavior and poor prognosis. Specifically, the expected tumor biology can be predicted by tumor histology and molecular markers, both of which are strongly associated with patient age.

## 2. Tumor Histology

Neuroblastoma tumor cells show varying degrees of differentiation that help predict patient prognosis. While neuroblastoma primarily contains immature cells, some have a component of fully mature ganglion cells that are typically found in a ganglioneuroma. A tumor with both elements of mature and immature cells is called a ganglioneuroblastoma. Tumors have been classified in detail according to this degree of differentiation by the International Neuroblastoma Pathology Committee [1,2,3] (Figure 1). Favorable and unfavorable histologic subtypes are based upon the level of Schwannian stroma present in the tumor, then further subclassified based upon the mitosis-karyorrhexis index (MKI) and patient age.

## 3. Molecular Markers

In addition to tumor histology, several genetic and chromosomal markers are strongly associated with tumor biology. The *MYCN* gene is perhaps the most important genetic marker of neuroblastoma aggressiveness. *MYCN* is an oncogene whose amplification is strongly associated with unfavorable clinical outcomes [4,5]. Another important prognostic marker in neuroblastoma is tumor cell ploidy. Neuroblastomas with triploidy or hyperdiploidy have been shown to have better outcomes than diploidy [5].

Segmental chromosomal anomalies with prognostic significance have also been identified for neuroblastoma. The most common are gain of 17q, loss of 1p, and loss of 11q, all of which are associated with a poorer prognosis [6,7]. Recent studies on familial neuroblastoma, which is rare, have also identified *ALK* and *PHOX2B* gene mutations. These mutations are found as germline mutations in patients with familial neuroblastoma but may also exist as somatic mutations in sporadic cases of neuroblastoma [8,9]. *ALK* aberrations in particular are now being factored into the new study protocols for treating high risk neuroblastoma. Finally, the presence of telomere-lengthening mechanisms appears to be associated with poorer prognosis as well as older patient age. Neuroblastoma telomere-lengthening can occur either via overexpression of the *TERT* gene, which encodes telomerase, or via mutation or deletion of the *ATRX* gene to activate the alternative lengthening of telomeres (ALT) pathway [10,11].

## 4. Current Risk Stratification

The International Neuroblastoma Risk Group (INRG) staging system stratifies patients based upon patient characteristics (namely age), disease presentation, and markers of tumor biology [12]. Unlike its predecessor the International Neuroblastoma Staging System (INSS), the INRG system is based entirely upon pretreatment tumor characteristics. A significant limitation of the INSS was that the stage for localized tumors depended on the extent of resection and lymph node sampling, which
(1)required that surgical resection be completed in order for staging to occur and(2)likely varied significantly amongst both surgeons and centers in terms of their judgment, skill, and aggressiveness towards complete resection and lymph node sampling.

On the other hand, the INRG staging system is independent of the completion or extent of surgical resection and lymph node sampling.

The risk groups of the INRG system are determined from the INRG stage, patient age, histology, *MYCN* gene amplification status, DNA ploidy status, and segmental chromosomal anomalies (11q aberration) (Figure 2). INRG stage (L1, L2, M, or MS) is determined radiologically based on the presence or absence of image-defined risk factors (IDRF) and metastatic disease [13] (Figure 3). IDRF essentially estimate the feasibility and safety of upfront surgical resection, with L1 tumors often amenable to resection and L2 tumors only rarely resectable at diagnosis (Figure 4). Encasement of a vessel is defined as greater than 50% of the circumference of the vessel being in contact with the tumor. A vein is also considered to be encased when it is flattened with no visible lumen present [14].

These risk factors (INRG stage, age, histology, and *MYCN*, 11q aberration, and ploidy status) in total are used to stratify patients into the following pre-treatment risk groups: very low, low, intermediate, and high (Figure 2). The importance of *MYCN* gene amplification status and patient age should be emphasized—all patients with *MYCN* amplification are classified as high risk, and older age (18 months being used as the cutoff) is strongly associated with worse outcomes.

## 5. Treatment by Risk Group

The risk groups determined from the INRG staging system are used to determine the optimal management strategy, with a focus on minimizing or avoiding treatment in low risk patients while intensifying treatment in high risk patients to improve survival. For the very low risk tumors, observation only is encouraged, as spontaneous regression is the norm. The current observation protocol, based upon a study by Nuchtern and colleagues [15], includes small (greatest tumor diameter <5 cm), non-infiltrative (INRG stage L1) tumors in children <12 months of age who are followed with serial ultrasounds and catecholamine studies. A 50% increase in either tumor volume or catecholamine levels triggers a move from observation to surgical resection. While the initial study included only adrenal tumors, the potential for spontaneous regression is thought to be at least as high if not higher for non-adrenal tumors [16]. Thus a Children’s Oncology Group (COG) study is currently under way to evaluate the feasibility of expanding the criteria for observation without surgery to include non-adrenal tumors.

For some low risk tumors (such as an INRG stage L1 tumor not meeting size requirements for observation), upfront surgical resection can be completed, with no or only minimal post-operative systemic therapy based upon the determined tumor biology. For other low risk and intermediate risk tumors (generally INRG stage L2 tumors without *MYCN* gene amplification), biopsy is performed, after which chemotherapy is initiated with or without surgical resection to follow. In general there has been a move towards less aggressive systemic and surgical therapy, given the safety of significantly reduced chemotherapy regimens [17] and the finding that unresected, residual, or recurrent neuroblastoma with favorable biology is likely to spontaneously regress and has little to no impact on survival [18,19].

Satisfactory treatment, through chemotherapy with or without surgery, is therefore considered to be a >50% reduction in tumor volume, or in some tumors with higher risk biology, a >90% reduction in tumor volume. An exception to this paradigm is INRG stage L2 tumors in children <18 months, in whom observation is encouraged for stable disease, moving to chemotherapy with or without surgery only when there is a >25% increase in tumor volume.

For high risk tumors, the first step in management is initiation of induction chemotherapy after obtaining tissue diagnosis. Induction chemotherapy regimens have some variation between centers but in general consist of about 6 cycles of a combination of platinum, alkylating, and topoisomerase-inhibitor agents. In addition, ^131^I-metaiodobenzylguanidine (^131^I-MIBG) is being utilized as initial therapy for some MIBG-avid neuroblastomas [20], and the *ALK* inhibitor crizotinib is being used for tumors with *ALK* aberrations (either *ALK* tyrosine kinase mutation or *ALK* mutation) [21]. A COG study is currently evaluating these new treatment options in greater detail—children with MIBG-avid high risk neuroblastoma are randomized to receive ^131^I-MIBG in addition to standard therapy, while children with MIBG-nonavid high risk neuroblastoma are non-randomly assigned to receive crizotinib in addition to standard therapy if *ALK* aberrations are identified.

During this period of induction chemotherapy, patients also undergo stem cell collection for later autologous stem cell transplantation. Surgical resection is undertaken near the end of induction chemotherapy, often after the 4th (typically), 5th, or 6th cycle. The goal of surgery is >90% resection, but en bloc resection of other organs (nephrectomy, pancreaticoduodenal resection, etc.) is to be avoided, and the aggressiveness of resection must be carefully weighed against the significant risks of morbidity and mortality.

Conflicting evidence exists regarding the importance of the extent of surgical resection for these high risk tumors. One study of INSS stage 4 patients >18 months old showed no significant difference in local control or survival between those who had resection of their primary tumor compared to those who did not [22]. On the other hand, other studies have shown improved outcomes with complete or near-complete resection of high risk tumors [23,24,25,26]. It is generally agreed upon from these studies that attempts should be made for >90% surgical resection while still minimizing the risk of any significant morbidity that would limit further systemic therapy.

Following surgical resection of high-risk tumors, any remaining induction chemotherapy is completed. The consolidation phase of treatment then begins, which is meant to eliminate remaining disease and consists of high dose chemotherapy followed by autologous stem cell transplantation (single or tandem) and radiation therapy. Finally, the post-consolidation or maintenance phase of therapy for high risk neuroblastoma begins, which is meant to prevent relapse and consists of isotretinoin in combination with anti-ganglioside 2 (GD2) antibody.

Finally, the special case of INRG stage MS disease (similar to the former INSS stage 4S) includes patients <18 months of age with L1 or L2 primary tumors, no unfavorable histologic or genetic features on biopsy, and metastases limited to the skin, liver, and bone marrow (no more than 10% involvement). In these patients, observation is recommended, with chemotherapy and/or surgical intervention reserved only for symptomatic patients.

## 6. Surgical Approach

For neuroblastomas requiring initiation of chemotherapy, tissue diagnosis has traditionally been obtained via an open biopsy. However, a minimally invasive approach (laparoscopy or thoracoscopy) is also widely used now. In addition, image-guided core needle biopsy is increasingly being used to provide a diagnosis and appears to provide similarly adequate results with potentially fewer complications [27,28,29].

Likewise, while open surgical resection has been the traditional approach, a minimally invasive surgical approach is now being used more frequently for neuroblastoma resection. For patients without IDRF, laparoscopic or thoracoscopic resection appears to provide a similarly adequate oncologic resection with potentially less blood loss, along with the typical benefits of less pain and quicker recovery [30,31,32,33,34]. For patients with IDRF, the feasibility of a minimally invasive approach is less clear. Open operations are still generally preferred, given the extensive tedious dissection required and the potential for rapid blood loss that would require prompt vascular control.

## 7. Conclusions

Much progress has been made in the prediction of neuroblastoma prognosis based upon tumor biology. This knowledge of tumor biology allows highly tailored therapy for children with neuroblastoma, from observation only to aggressive multimodal systemic therapy and surgery. This approach, along with additional discoveries to come, can focus intense therapy on children who need it while avoiding unnecessary treatment in those who do not in order to maximize outcome while simultaneously minimizing unnecessary morbidity.

## Figures and Tables

**Figure 1 children-06-00012-f001:**
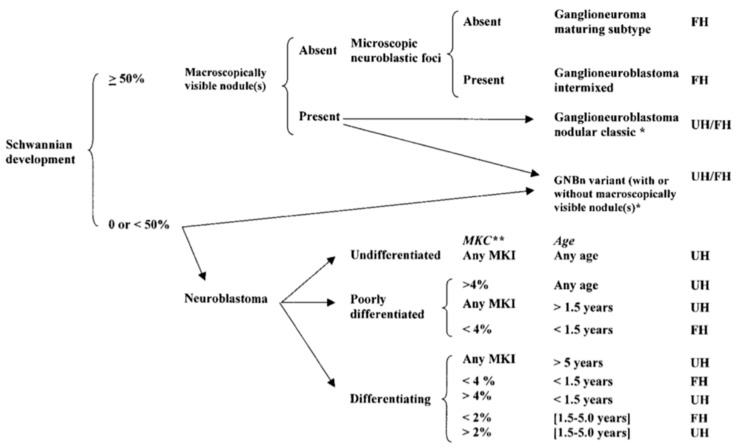
International Neuroblastoma Pathology Classification [3] (Reprinted with permission from Peuchmaur M. et al.: *Cancer* 98(10): 2274–2281. © 2003 American Cancer Society. All rights reserved.). FH: favorable histology; UH: unfavorable histology. GNBn—ganglioneuroblastoma, nodular; MKC—mitotic and karyorrhectic cells; MKI—mitosis-karyorrhexis index; **: 2% = 100 of 5000 cells, 4% = 200 of 5000 cells.

**Figure 2 children-06-00012-f002:**
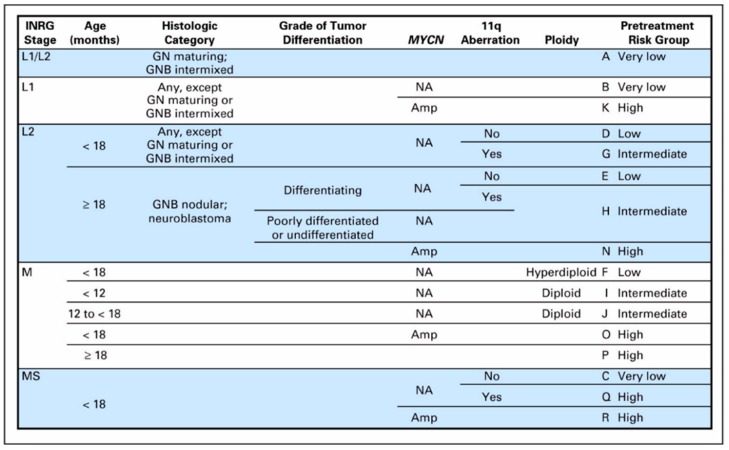
International Neuroblastoma Risk Group (INRG) pre-treatment classification system [12] (Reprinted with permission from Cohn S. et al.: *J. Clin. Oncol.* 27(2): 289–297. © 2009 American Society of Clinical Oncology. All rights reserved.). GN—ganglioneuroma; GNB—ganglioneuroblastoma; Amp—amplified; NA—not amplified; blank fields represent any value.

**Figure 3 children-06-00012-f003:**
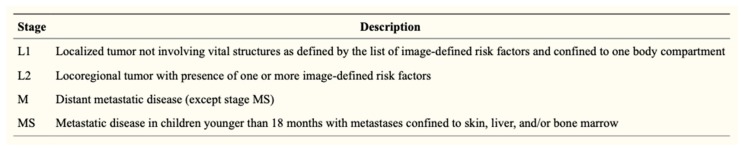
INRG staging system [13] (Reprinted with permission from Monclair T. et al.: *J. Clin. Oncol.* 27(2): 298–303. © 2009 American Society of Clinical Oncology. All rights reserved.).

**Figure 4 children-06-00012-f004:**
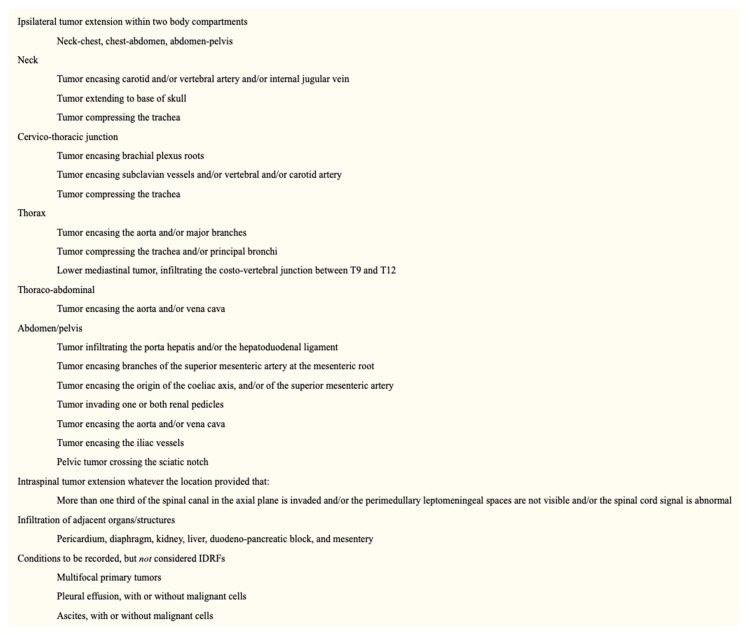
Image-defined risk factors for the INRG classification system [13] (Reprinted with permission from Monclair T. et al.: *J. Clin. Oncol.* 27(2): 298–303. © 2009 American Society of Clinical Oncology. All rights reserved.).

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
