# Peer review of "Neuroblastoma: Tumor Biology and Its Implications for Staging and Treatment"

_children, 2019, doi:10.3390/children6010012_

Round 1

Reviewer 1 Report

This review paper is a well written manuscript which is a very nice summary of the current clinical management of neuroblastoma. Importantly, the authors focus the readership on the current INRG classification and discuss the new and important role of IDRFs.  A nice discussion of the current surgical management and the relevant literature is included.

Author Response

We appreciate these kind words and enthusiasm for our review.

Reviewer 2 Report

Well-written paper summarising the already well-known data on the biology and risk group treatment of neuroblastoma. The article is didactic for people who don't know about neuroblastoma but brings nothing new on the field of neuroblastoma.

Author Response

We appreciate the enthusiasm for our review. We agree that as a review paper this manuscript is a summary of existing data and does not present new findings.

Reviewer 3 Report

This is a well-written, concise manuscript detailing neuroblastoma tumor biology and its impact on treatment. Overall, the manuscript is acceptable for publication. I feel that the following items for discussion should be added to the manuscript to make it more complete for the reader.

1.       Because ALK aberrations are being factored into the treatment grouping for high-risk neuroblastoma under the new COG protocol ANBL1531 (Iobenguane I-131 or Crizotinib and Standard Therapy in Treating Younger Patients with Newly-Diagnosed High-Risk Neuroblastoma or Ganglioneuroblastoma), I feel that the discussion of ALK aberrations should be more complete. ALK aberrations may come in the form of tyrosine kinase receptor mutations or ALK amplification in neuroblastoma. Patients with either of these aberrations are being non-randomly assigned to receive criotinib (an Alk inhibitor) in this protocol in addition to the current COG-recommended high-risk therapy that is outlined in the manuscript. It should be noted that this is a treatment option under clinical evaluation, but I feel it is important to discuss in the manuscript.

2.       Because MIBG avidity is also being factored into the treatment grouping for high-risk neuroblastoma under ANBL1531, I feel that the discussion of 131I-MIBG therapy should be more complete. A concise review of ANBL1531 can be accessed on pubmed.gov under the Neuroblastoma Treatment (PDQ): Health Professional Version. For the author’s review, patients with MIBG-avid, ALK wild-type (or ALK unknown) disease will be randomly assigned to one of the following three arms under ANBL1531:

Current COG-recommended       high-risk therapy, including four more cycles of induction chemotherapy       and surgical resection of the primary tumor, consolidation with tandem       transplant and focal external-beam radiation therapy, and dinutuximab       immunotherapy with isotretinoin.

Current COG-recommended       high-risk therapy with the addition of a block of 131I-MIBG after the       third induction cycle.

Current COG-recommended       high-risk therapy with the addition of a block of 131I-MIBG after the       third induction cycle and the substitution of busulfan/melphalan single       autologous SCT in place of tandem transplant.

 It should be noted that this is a treatment option under clinical evaluation, but I feel it is important to discuss in the manuscript.

3.        In the section Treatment by Risk Group, the study by Nuctern et. al. is mentioned which supports observation alone without biopsy in infants < 12 months of age who have small adrenal masses. Under ANBL1232 (Response and Biology-Based Risk Factor–Guided Therapy in Treating Younger Patients With Non–High-Risk Neuroblastoma), expanded observational criteria are being clinically evaluated.  Observation without biopsy is now being evaluated for patients with nonadrenal primary tumors. Table 9 under Treatment of Low-Risk Neuroblastoma in the PDQ document mentioned above summarizes the observational approach currently under clinical evaluation. It should be noted that this is a treatment option under clinical evaluation, but I feel it is important to discuss or at least mention that these criteria are being evaluated for expansion in the manuscript.

4.       Since this article is being published in a surgical journal, I feel that the discussion of imaging defined risk factors should be more complete as it pertains to the relationship between the primary tumor and vital structures. Specific issues that should be discussed include the percent contact between the tumor and arterial vessel to quality as an IDRF and, for venous structures, the visibility of the lumen to qualify as an IDRF.  These can be reviewed in the article Guidelines for Imaging and Staging of Neuroblastic Tumors: Consensus Report from the International Neuroblastoma Risk Group Project by Herve Brisse et. al in Radiology Vol 261(1) 2011.

5.       Figure 1 has two asterixes (*) that are not explained in the figure legend.

6.       Consider mentioning the impact of telomere lengthening mechanisms on neuroblastoma prognosis. 1. TERT/telomerase mediated mechanisms and 2. ATRX mutation/deletion and its association with alternative telomere lengthening and older age neuroblastoma/poor prognosis.

Author Response

This is a well-written, concise manuscript detailing neuroblastoma tumor biology and its impact on treatment. Overall, the manuscript is acceptable for publication.

We appreciate the enthusiasm for our review, as well as the many helpful suggestions outlined below.

I feel that the following items for discussion should be added to the manuscript to make it more complete for the reader.

1. Because ALK aberrations are being factored into the treatment grouping for high-risk neuroblastoma under the new COG protocol ANBL1531 (Iobenguane I-131 or Crizotinib and Standard Therapy in Treating Younger Patients with Newly-Diagnosed High-Risk Neuroblastoma or Ganglioneuroblastoma), I feel that the discussion of ALK aberrations should be more complete. ALK aberrations may come in the form of tyrosine kinase receptor mutations or ALK amplification in neuroblastoma. Patients with either of these aberrations are being non-randomly assigned to receive criotinib (an Alk inhibitor) in this protocol in addition to the current COG-recommended high-risk therapy that is outlined in the manuscript. It should be noted that this is a treatment option under clinical evaluation, but I feel it is important to discuss in the manuscript.

This is an excellent point, and we have added discussion regarding the current protocol that includes treatment with ALK inhibitor.

2. Because MIBG avidity is also being factored into the treatment grouping for high-risk neuroblastoma under ANBL1531, I feel that the discussion of 131I-MIBG therapy should be more complete. A concise review of ANBL1531 can be accessed on pubmed.gov under the Neuroblastoma Treatment (PDQ): Health Professional Version. For the author’s review, patients with MIBG-avid, ALK wild-type (or ALK unknown) disease will be randomly assigned to one of the following three arms under ANBL1531:

Current COG-recommended high-risk therapy, including four more cycles of induction chemotherapy and surgical resection of the primary tumor, consolidation with tandem       transplant and focal external-beam radiation therapy, and dinutuximab  immunotherapy with isotretinoin.

Current COG-recommended high-risk therapy with the addition of a block of 131I-MIBG after the third induction cycle.

Current COG-recommended high-risk therapy with the addition of a block of 131I-MIBG after the third induction cycle and the substitution of busulfan/melphalan single  autologous SCT in place of tandem transplant.

It should be noted that this is a treatment option under clinical evaluation, but I feel it is important to discuss in the manuscript.

Thank you for the excellent suggestions. We have added more detailed discussion of MIBG therapy and its inclusion in the current study protocol.

3. In the section Treatment by Risk Group, the study by Nuchtern et. al. is mentioned which supports observation alone without biopsy in infants < 12 months of age who have small adrenal masses. Under ANBL1232 (Response and Biology-Based Risk Factor–Guided Therapy in Treating Younger Patients With Non–High-Risk Neuroblastoma), expanded observational criteria are being clinically evaluated. Observation without biopsy is now being evaluated for patients with nonadrenal primary tumors. Table 9 under Treatment of Low-Risk Neuroblastoma in the PDQ document mentioned above summarizes the observational approach currently under clinical evaluation. It should be noted that this is a treatment option under clinical evaluation, but I feel it is important to discuss or at least mention that these criteria are being evaluated for expansion in the manuscript.

This is an excellent point and we have added discussion to the manuscript regarding the study on expanded observational criteria.

4. Since this article is being published in a surgical journal, I feel that the discussion of imaging defined risk factors should be more complete as it pertains to the relationship between the primary tumor and vital structures. Specific issues that should be discussed include the percent contact between the tumor and arterial vessel to quality as an IDRF and, for venous structures, the visibility of the lumen to qualify as an IDRF. These can be reviewed in the article Guidelines for Imaging and Staging of Neuroblastic Tumors: Consensus Report from the International Neuroblastoma Risk Group Project by Herve Brisse et. al in Radiology Vol 261(1) 2011.

We have added more detailed discussion of image-defined risk factors and have added the suggested reference.

5.  Figure 1 has two asterixes (*) that are not explained in the figure legend.

Thank you for noting this omission. It has been corrected.

6.  Consider mentioning the impact of telomere lengthening mechanisms on neuroblastoma prognosis. 1. TERT/telomerase mediated mechanisms and 2. ATRX mutation/deletion and its association with alternative telomere lengthening and older age neuroblastoma/poor prognosis.

We have added a brief discussion of telomere lengthening mechanisms as recommended.

Round 2

Reviewer 2 Report

The authors improved the manuscript by adding the reviewers' suggestions.